# Difensil Immuno Reduces Recurrence and Severity of Tonsillitis in Children: A Randomized Controlled Trial

**DOI:** 10.3390/nu12061637

**Published:** 2020-06-02

**Authors:** Arianna Di Stadio, Antonio della Volpe, Fiammetta M. Korsch, Antonietta De Lucia, Massimo Ralli, Francesco Martines, Giampietro Ricci

**Affiliations:** 1 Department of Otolaryngology, University of Perugia, 06129 Perugia, Italy; ricci1501@hotmail.com; 2 Otology and Cochlear Implant Unit, Santobono-Pausilipon Children’s Hospital, 80129 Naples, Italy; antonidellavolpe@yahoo.it (A.d.V.); fiammettakorsch@gmail.com (F.M.K.); toniadelucia@live.it (A.D.L.); 3 Sense Organs Department, Sapienza University of Rome, 00161 Rome, Italy; massimo.ralli@uniroma1.it; 4 Biomedicine, Neuroscience and Advanced Diagnostics Department, University of Palermo, 90127 Palermo, Italy; francesco.martines@unipa.it

**Keywords:** tonsillitis, treatment, oral supplement, immune system, immune stimulation

## Abstract

Oral supplements (OS) support the immune system in fighting upper airways infection. This study aimed to analyze the effect of Difensil Immuno (DI) on the recurrence of tonsillitis and fever in children. A multicentric randomized clinical trial was conducted. One-hundred and twenty children with chronic tonsillitis were randomly assigned to group A, B or control. Patients in group A were treated with 10 mL of DI for 90 consecutive days, patients in group B underwent treatment with 15 mL of DI for 45 consecutive days. The following data were collected at baseline (T0), T1 and T2: tonsillitis and fever episodes, tonsillar volume, blood test results. One-way ANOVA was used to analyze within and between variances. Patients in group A and B statistically improved their clinical parameters (episode of tonsillitis and fever, tonsillar volume) when compared to control group both at T1 and T2. However, T1 variances were more consistent in group A than in group B. All patients in the study groups improved their clinical outcomes. No statistically significant variances were observed in blood parameters both at T1 and T2. Our results suggest that children treated with DI had fewer episodes of tonsillitis and fever and a reduction in their tonsillar volume.

## 1. Introduction

The benefic effect of oral supplements (OS) on the immune system has been shown in patients suffering from immunodeficiency [1] and in subjects with cancer [2,3]; some specific vitamins, such as vitamin D, were shown to be able to immunomodulate the immune system of patients with Multiple Sclerosis by reducing neurodegeneration [4].

Vitamins and antioxidant molecules act on the immune system in different ways; for example, some of them increase the number of circulating white cells [5], while others empower the reactivity of the immune response by reducing the level of reactive oxygen species (ROS) known to be dangerous for the immune system [6].

Most studies with OS are conducted in the adult population and studied these compounds in non-infectious diseases [1,2,3]. On the contrary, a few studies have been performed on children suffering from infectious disease to test the efficacy of OS [7].

Recently, a randomized clinical trial from our group showed that an OS containing Sambucus nigra, zinc, tyndallized Lactobacillus acidophilus (HA122), arabinogalactans, vitamin D, vitamin E and vitamin C (commercial name Difensil Immuno (DI), available in Italy and Europe) could improve the outcome of children with otitis media [8]. In this study, DI improved the immune response and reduced the virulence of infection [8].

Each component of DI interacts differently with the immune system. Sambucus nigra inhibits viral replication and increases the production of inflammatory cytokines; vitamin C reduces the circulating oxidative species, whose presence negatively impact on the immune response; vitamin D promotes differentiation of monocytes/macrophages in their active form and increases the chemotactic and phagocytic capacity of these cells; zinc improves the macrophage phagocytosis ability, the capacity of antigen presentation and signal transmission between these cells and the other belonging to the immune-system; selenium is a potent immune-stimulant able to improve T cell proliferation, natural killer cell activity and several innate immune cell functions. Finally, Lactobacillus acidophilus and arabinogalactans stimulate the macrophage activity and the production of cytokines.

Children commonly suffer from tonsillitis due to their immature immune system; generally, antibiotics and corticosteroids are used for the treatment of recurrent episodes [9]. The aim of this randomized clinical trial was to analyze the effect of DI on the recurrence of tonsillitis and fever in children.

## 2. Materials and Methods

This multicentric study was conducted in the Otolaryngology departments of the following institutions: Santobono-Pausilipon hospital, University of Perugia and University of Palermo from October 2019 to March 2020. All procedures were approved by the local Institutional Review Board committee of each hospital and were conducted in accordance with the ethical principles outlined in the Declaration of Helsinki. The parents of the participating children signed a written informed consent document authorizing the enrollment of their child in the study.

The three centers used the same criteria and the procedures were standardized before starting the study.

The inclusion criteria were age <8, at least 3 episodes of tonsillitis per year with fever, Mackenzie score >2, no previous history of adenoidectomy, no current treatment with antibiotics, no history of allergies, no diabetes or severe neurological disease.

In each hospital, patients were randomly assigned by a computer to one of three groups: Group A (GA), Group B (GB) and control Group (CG). GA and GB were treated with OS with immune-stimulating molecules (Sambucus nigra, zinc, tyndallized Lactobacillus acidophilus (HA122), arabinogalactans, vitamin D, vitamin E and vitamin C) (Difensil Immuno, Humana Italia S.p.A.).

GA included patients who were treated with 10 mL of DI for 90 consecutive days; GB patients underwent treatment with 15 mL of DI for 45 consecutive days; and lastly CG did not receive any therapy. DI was always administered in a single dose in the morning during breakfast.

A dose of 10 mL of DI contains 112 mg of Sambucus nigra, 7.5 mg of zinc, 1 × 10^7^ tyndallized Lactobacillus acidophilus (HA122), 10 mg of arabinogalactans, 10 mcg of vitamin D, 30 mg of vitamin E and 90 mg of vitamin C. A dose of 15 mL of DI contains 183 mg of Sambucus nigra, 11.2 mg of zinc, 1.5 × 10^7^ tyndallized Lactobacillus acidophilus (HA122), 15 mg of arabinogalactans, 15 mcg of vitamin D, 45 mg of vitamin E and 135 mg of vitamin C.

In cases where fever exceeded 38 °C, children were treated with paracetamol (500 mg) and antibiotics if fever persisted over three days.

Patients in the CG followed a “wait and see approach”; children were treated with anti-inflammatory drugs in cases of a fever episode over 38 °C, and antibiotics if the condition did not resolve within three days. Corticosteroids were never used.

Outcome measures included: number of episodes of tonsillitis during the observation period and episodes of fever during the period of follow-up, inspection of tonsil aspect and classification following the Mackenzie classification (Figure 1), red cell count, white cell count, hemoglobin (Hb), polymerase chain reaction (PCR) value, erythrocyte sedimentation rate (ESR), antistreptolysin O titer (ASLOT), pro-calcitonin, rheumatologic test (RT).

Three time points were identified: T0 = before treatment (baseline), T1 = 45 days after treatment and T2 = 90 days after treatment; at each follow-up all outcome measures were collected.

Episodes of tonsillitis and fever at T0 were counted by considering the last 3 months (90 days) before the beginning of the study.

One-hundred twenty subjects were enrolled; each center recruited 40 patients. In detail, Group A included 40 children, average age = 6 years (SD: 2.6; CI 95%: 3–12); 25 were females and 15 males. Group B included 41 children, average age = 6 years (SD: 2.5; CI: 3–12), 22 were females and 19 males. Control group included 39 children, average age = 5.6 years (SD: 2.3; CI 95%: 3–12), 20 females and 19 males. None of the patients underwent antibiotic therapy, 15 days being included in the study for all the observation period up to one week after the end of the study.

All patients completed the study and none of them dropped or missed follow-up visits.

### Statistical Analysis

The statistical analysis was performed using STATA^®^. One-way ANOVA was used to evaluate the scores variation within each group (GA, GB and CG) at three time points (T0, T1 and T2) for the number of episodes of tonsillitis and fever. The same test was repeated to evaluate the variance of the tonsils (Mackenzie) and the blood parameters. A Bonferroni-Holms ad hoc test was performed for each one-way ANOVA. A *p* value < 0.05 was considered statistically significant.

## 3. Results

### 3.1. Treatment Results “within” Group Comparison

#### 3.1.1. Group A

None of the patients needed antibiotics treatment (Figure 2 and Table 1).

Statistically significant variations were observed by comparing T0, T1 and T2 follow-up for the following parameters:(1)Number of tonsillitis episode (ANOVA: *p* < 0.0001) between T0 and T1 (BH: *p* < 0.0001), between T0 and T2 (BH: *p* < 0.0001) and T1 and T2 (BH: *p* < 0.0001).(2)Mackenzie score (ANOVA: *p* < 0.0001). In particular a significant variance was observed between T0 and T1 (HB: *p* < 0.0001), T0 and T2 (HB: *p* < 0.0001). No statistically significant variances were observed between T1 and T2 (HB: *p* = 0.3).(3)Number of fever episode (ANOVA: *p* < 0.0001). The variance was statistically significant by comparing T0 and T1 (HB: *p* < 0.0001), T0 and T2 (HB: *p* < 0.0001) and T1 and T2 (HB: *p* < 0.0001).

No statistically significant variations were observed by comparing children at T0, T1 and T2 for the following parameters: red cell count (ANOVA: *p* = 1), white cell count (ANOVA: *p* = 1), Hb (ANOVA: *p* = 1), PCR value (ANOVA: *p* = 0.6), ESR (ANOVA: *p* = 0.4), ASLOT (ANOVA: *p* = 1), pro-calcitonin (ANOVA: *p* = 0.9), RT (ANOVA: *p* = 1).

#### 3.1.2. Group B

None of the patients were treated with antibiotics (Figure 2 and Table 2).

Statistically significant variations were observed by comparing T0, T1 and T2 follow-up for the following parameters:(1)Number of tonsillitis episode (ANOVA: *p* < 0.0001) between T0 and T1 (BH: *p* < 0.0001), between T0 and T2 (BH: *p* < 0.0001). No statistically significant variances were observed between T1 and T2 (HB: *p* = 0.07).(2)Mackenzie score (ANOVA: *p* < 0.0001). In particular, a significant variance was observed between T0 and T2 (HB: *p* < 0.0001) and between T1 and T2 (HB: *p* = 0.01). No statistically significant variances were observed between T0 and T1 (HB: *p* = 0.06).(3)Number of fever episode (ANOVA: *p* = 0.002). The variance was statistically significant by comparing T0 and T1 (HB: *p* < 0.01), T0 and T2 (HB: *p* < 0.01) and T1 and T2 (HB: *p* < 0.01).

No statistically significant variations were observed by comparing children at T0, T1 and T2 for the following parameters: red cell count (ANOVA: *p* = 1), white cell count (ANOVA: *p* = 1), Hb (ANOVA: *p* = 1), PCR value (ANOVA: *p* = 0.9), ESR (ANOVA: *p* = 0.5), ASLOT (ANOVA: *p* = 1), pro-calcitonin (ANOVA: *p* = 0.9), RT (ANOVA: *p* = 1).

#### 3.1.3. Control Group

Fifty per-cent of children needed antibiotics treatment (Figure 2 and Table 3).

No statistically significant variations were observed by comparing children at T0, T1 and T2 for the following parameters: number tonsillitis episodes between the three follow-up (ANOVA: *p* = 0.2), Mackenzie scores (ANOVA: *p* = 0.9), fever episodes (ANOVA: *p* = 0.2) red cell count (ANOVA: *p* = 1), white cell count (ANOVA: *p* = 1), Hb (ANOVA: *p* = 1), PCR value (ANOVA: *p* = 0.6), ESR (ANOVA: *p* = 0.9), ASLOT (ANOVA: *p* = 1), pro-calcitonin (ANOVA: *p* = 0.7), RT (ANOVA: *p* = 1).

### 3.2. Treatment Results “between” Group Comparison

Statistically significant variances were observed between GA, GB and CG for the following parameters:

#### 3.2.1. Comparison at T1

(1)Number of tonsillitis episode (ANOVA: *p* < 0.0001). We observed statistically significant variances between GA (average: 3.9; SD: 1; CI 95%: 2–5) and CG (average: 4.1; SD: 0.9; CI 95%: 2–5) (BH: *p* < 0.0001), but not between GB (average: 4; SD: 1.1; CI 95%: 2–7) and CG (BH: *p* > 0.05). Statistically significant variances were observed between GA and GB (HB: *p* < 0.0001).(2)Mackenzie score (ANOVA: *p* < 0.0001). GA (average: 2.5; SD: 1; CI 95%: 1–4) showed statistically significant variances with CG (average: 3; SD: 0.6; CI 95%: 2–4) (HB: *p* = 0.0007), while no statistically significant variances were observed between GB (average: 3; SD: 0.6; CI 95%: 2–4) and CG (HB: *p* > 0.05). Statistically significant variance was observed by comparing GA and GB (HB: *p* < 0.00001).(3)Number of fever episode (ANOVA: *p* = 0.002). The variance was statistically significant by comparing GA (average: 2; SD: 0.7; CI 95%: 1–3) with CG (average: 2.4; SD: 0.7; CI 95%: 1–4) (HB: *p* = 0.001) and GB (average: 2.3; SD: 0.7; CI 95%: 1–4) with CG (HB: *p* = 0.002). No statistically significant differences were observed between GA and GB.

No statistically significant variations were observed by comparing GA, GB and CG at T1 for the following parameters: red cell count (ANOVA: *p* = 1), white cell count (ANOVA: *p* = 1), Hb (ANOVA: *p* = 1), PCR value (ANOVA: *p* = 0.7), ESR (ANOVA: *p* = 0.7), ASLOT (ANOVA: *p* = 0.9), pro-calcitonin (ANOVA: *p* = 0.9), RT (ANOVA: *p* = 1) (Figure 3).

#### 3.2.2. Comparison at T2

(1)Number of tonsillitis episode (ANOVA: *p* < 0.0001). By comparing GA (average: 2.8; SD: 0.8; CI 95%: 1–4) and CG (average: 4.1; SD: 1; CI 95%: 2–6) we observed statistically significant variances at T2 (HB: *p* < 0.0001) and the same results were observed when comparing GB (average: 3.3; SD: 0.9; CI 95%: 2–5) with CG (HB: *p* = 0.0002). No statistically significant variances were observed between GA and GB.(2)Mackenzie score (ANOVA: *p* < 0.0001). GA (average: 2.3; SD: 0.7; CI 95%: 1–3) showed statistically significant variances with CG (average: 3; SD: 0.6; CI 95%: 2–4) at T2 (HB: *p* = 0.0001), as well as GB (average: 2.5; SD: 0.5; CI 95%: 2–3) versus CG (HB: *p* = 0.01). A statistically significant variance was observed by comparing GA and GB (HB: *p* = 0.003).(3)Number of fever episode (ANOVA: *p* = 0.002). The variance was statistically significant by comparing GA (average: 1.3; SD: 0.6; CI 95%: 0–2) and CG (average: 2.2; SD: 0.9; CI 95%: 1–3) at T2 (HB: *p* = 0.001) and GB (average: 1.7; SD: 0.7; CI 95%: 1–3) and CG at T2 (HB: *p* = 0.002). No statistically significant differences were observed between GA and GB.

No statistically significant variations were observed by comparing GA, GB and CG at T2 for the following parameters: red cell count (ANOVA: *p* = 1), white cell count (ANOVA: *p* = 1), Hb (ANOVA: *p* = 1), PCR value (ANOVA: *p* = 0.7), ESR (ANOVA: *p* = 0.7), ASLOT (ANOVA: *p* = 0.9), pro-calcitonin (ANOVA: *p* = 0.9), RT (ANOVA: *p* = 1) (Figure 3).

## 4. Discussion

Our results show that DI reduced the recurrence of tonsillitis and the fever episodes over a period of three months and reduced the average volume of tonsils. Furthermore, children who used DI never needed antibiotics treatment. Despite these clinical results, the use of OS did not modify the blood parameters of inflammation (PCR, ESR and pro-calcitonin). However, the supplement did not affect the other blood parameters.

All patients (“between analysis”), independently from the therapeutic scheme used, showed a better outcome than CG, supporting the use of DI as a stimulator of natural immune answers in children.

Patients treated with 10 mL of OS for 90 consecutive days (GA) showed an improvement after 45 days, while patients who underwent 15 mL × 45 days (GB) significatively improved their outcome after 90 days. This result showed that the use of a low dosage for a longer period could reduce the episode and the severity of tonsillitis better than a higher dosage for a short time, as has already been shown for other types of OS [10].

On the other hand, patients in GB were able to improve their condition after the end of the treatment; it could be speculated that a high dosage, despite less efficiency in the short time, could be released slowly [11] and improve patients’ outcomes after treatment suspension.

The “within” analysis group partially confirmed the observations of the “between groups” analysis; in fact, both therapeutic schemes improved patients’ outcomes.

Patients in GA showed improvements in all outcomes (episode of tonsillitis, Mackenzie score and fever episode) after 45 days of treatment and this improvement was consistent at the second follow-up (90 days); children in GB presented a reduced number of tonsillitis cases and fever episodes at T1 without a statistically significant change in the tonsil volume; this outcome improved at T2. In addition, patients in GB showed a further reduction of tonsillitis and fever episodes after 90 days, by reaching results similar to those observed in GA.

We speculate that the delay in tonsil volume reduction in GB could be related to the high concentration of DI, which stimulates an increase in the local immune response [12] with consequent persistence of tonsillar hypertrophy. This initial over-stimulation of white cells in the Waldeyer ring could explain the improvement observed in these patients after the end of oral supplementation. We previously showed the efficacy of this specific OS to improve the immune answer in the Waldeyer ring and to determine an important improvement of otitis media in children [8].

Overall, our results suggest that a long-term treatment with 10 mL of OS is better than a shorter one, although the short time, high-dosage therapeutic scheme maintains a very good efficacy after treatment suspension.

We speculate that DI could have stimulated the immune system function [13,14,15,16] thanks to the synergic effect of its different components. *S. nigra*, which has been confirmed as an efficient molecule for inhibiting viral replication in the early stages of infection [17] was fundamental to reduce the episode of tonsillitis and consequently to decrease fever recurrence. A, C, D and E vitamins contributed to the action of the *S. nigra* by increasing mucosal IgA (A and E) [18,19] and by actively stimulating macrophage and lymphocyte activity [5,20], thus making children more resistant to viral infection in the upper respiratory tract (URT). The reduction of the number of infections progressively allowed the reduction of tonsil volume.

Lactobacillus acidophilus, another component of the OS, can improve the immune response thanks to the reduction of concentration of ROS, known as a depressor of the immune system [6]. In addition, lactobacillus inhibits the penetrance of virus inside the cells by further contributing to protect URT from common viral infection [21] and has the capacity to inhibit the adhesion and the growth of gram-negative bacteria [22] by protecting the subject from viral and bacterial aggression.

The results of immune-stimulating OS treatment in children reported in this study are consistent with the results observed in adults [23,24,25] and with a previous study conducted in children in which the authors showed the efficacy of this OS due to its ability to reduce the volume of the adenoid tissue indicative of the resolution of the infective/inflammatory process [26].

However, because of the short duration of this study, additional studies with a longer follow-up are necessary to confirm these preliminary results. In fact, longer studies could evaluate the effect of DI on immune-system cells and could evaluate whether DI could be tolerated for longer periods without interruption.

## 5. Conclusions

Our study results suggest that OS can be a valid tool to reduce the recurrence of tonsillitis and fever episodes in children. In our patients, this natural compound showed a positive effect on the chronic disease without side effects. Although this is the first clinical trial conducted on children with recurrent tonsillitis to evaluate the efficacy of DI, we already successfully tested this compound on children with otitis media.

## Figures and Tables

**Figure 1 nutrients-12-01637-f001:**
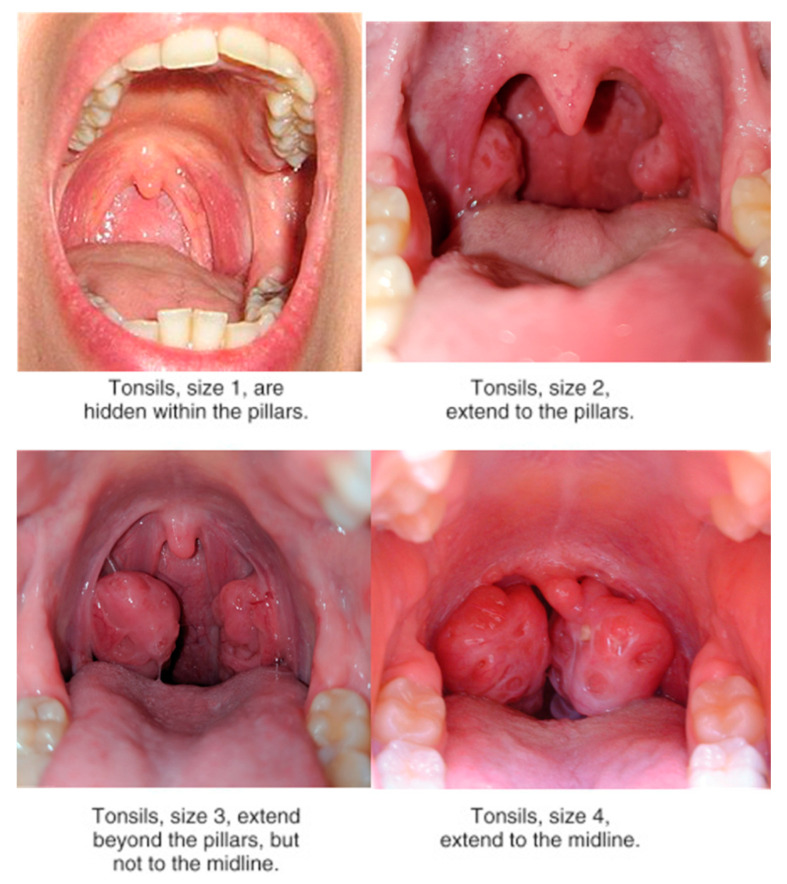
The image shows details of the Mackenzie classification. We used pictures from our sample to detail the differences between different severities of tonsillar aspect.

**Figure 2 nutrients-12-01637-f002:**
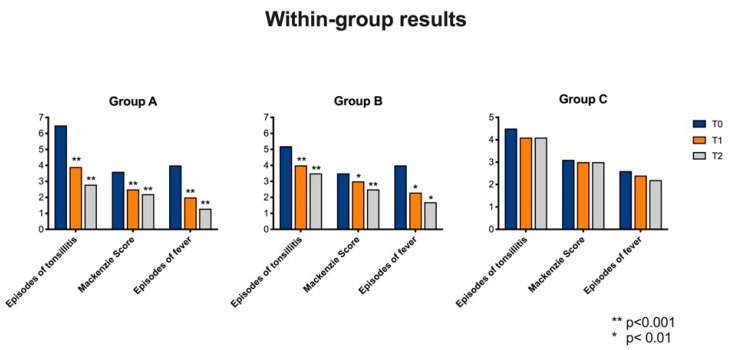
Variation within each group by comparing T0, T1 and T2 for episodes of tonsillitis, Mackenzie Score and episodes of fever.

**Figure 3 nutrients-12-01637-f003:**
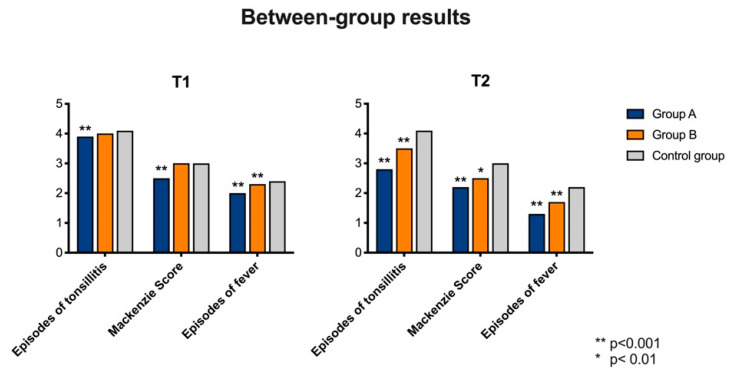
Differences of the analyzed variables between Group A, Group B and Control Group. Patients in Group A obtained a statistically significant improvement from T0 as detailed in the results.

**Table 1 nutrients-12-01637-t001:** Average of clinical data in Group A observed at the three different follow-up visits.

Group A	T0	T1	T2
*Episodes of tonsillitis*	6.5 (SD: 1; CI 95%:4–8)	3.9 (SD: 1; CI 95%:2–5)	2.8 (SD: 0.9; CI 95%:1–4)
*Mackenzie Score*	3.6 (SD: 0.5; CI 95%:3–4)	2.5 (SD: 1; CI 95%:1–4)	2.2 (SD: 0.7; CI 95%: 1–3)
*Episodes of fever*	4 (SD: 1 CI 95%: 4–8)	2 (SD: 0.7; CI 95%: 1–3)	1 (SD: 0.6; CI 95%: 0–2)

**Table 2 nutrients-12-01637-t002:** Average of clinical data in Group B observed at the three different follow-up visits.

Group B	T0	T1	T2
*Episodes of tonsillitis*	5,2 (SD: 1.2 CI 95%: 4–8)	4 (SD: 1.1; CI 95%: 2–7)	3.3 (SD: 0.9; CI 95%: 2–5)
*Mackenzie Score*	3.5 (SD: 0,6 CI 95%:2–4)	3 (SD: 0.6; CI 95%:2–4)	2.5 (SD: 0.5; CI 95%:2–4)
*Episodes of fever*	4 (SD 0.1 CI 95%: 4–8)	2,3 (SD: 0.7; CI 95%:1–3)	1.7 (SD: 0.8; CI 95%:1–3)

**Table 3 nutrients-12-01637-t003:** Average of clinical data in group B observed at the three different follow-up visits.

Control Group	T0	T1	T2
*Episodes of tonsillitis*	4.5 (SD: 0.9 CI 95%: 3–6)	4.1 (SD: 0.9 CI 95%:2–5)	4.1 (SD: 1; CI 95%:2–6)
*Mackenzie Score*	3 (SD: 0.5 CI 95%:2–4)	3 (SD: 0.6 CI 95%:2–4)	3 (SD: 0.6 CI 95%:2–4)
*Episodes of fever*	2.6 (SD: 0.8 CI 95%: 4–8)	2.4 (SD: 0.6 CI 95%:2–4)	2.2 (SD: 0.8 CI 95%:1–4)

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
