# Peer review of "Difensil Immuno Reduces Recurrence and Severity of Tonsillitis in Children: A Randomized Controlled Trial"

_nutrients, 2020, doi:10.3390/nu12061637_

Round 1

Reviewer 1 Report

The idea of evaluating supplements in children and analyzing the relationship with the immune system is very interesting. The methodology of the article is well described and the statistical analyzes are very clear. However, some suggestions for improving the presentation and understanding of the article are presented below.

  • The supplements covered in the articles (vitamin C, E zinc .... must be present in the introduction). It should be mentioned how these elements can interact with the immune system.
  • Line 54. Insert the protocol number of the study approval after submission to the ethics committee
  • The statistical data presented in the results (for example item 3.1.1 group A; 3.1.2 group B and 3.1.3 group C) can be better presented in a table.
  • In the opinion of the authors, can the supplements analyzed in the article be suggested as supplements in the form of drugs for children of this age in the medical routine?

Author Response

Dear Reviewer 1

Thanks for reviewing our manuscript and for the valuable comments that helped us clarify some relevant aspects that were missed or unclear in the first version of the paper. We hope that the changes made in the revised manuscript and responses provided below have adequately addressed the reviewers’ comments and made this paper stronger.

The idea of evaluating supplements in children and analyzing the relationship with the immune system is very interesting. The methodology of the article is well described and the statistical analyzes are very clear. However, some suggestions for improving the presentation and understanding of the article are presented below.

1) The supplements covered in the articles (vitamin C, E zinc .... must be present in the introduction). It should be mentioned how these elements can interact with the immune system.

  • Thanks for reviewing our manuscript and for the valuable comments that helped us clarify some relevant aspects that were missed or unclear in the first version of the paper. Following your suggestion, we added a paragraph in the introduction that specified the effect of the different molecules on the immune system

2) Line 54. Insert the protocol number of the study approval after submission to the ethics committee.

  • We have not specified the protocol number in the study because all procedures were specifically approved by the ethical committees of the involved centers without the release of a protocol number. This is the standard procedure in our country for studies that involve natural compounds, and require only a generic authorization of the ethical committee given that all procedures for patient safety are respected. Written informed consent has been provided by the parents of all participating children.

3) The statistical data presented in the results (for example item 3.1.1 group A; 3.1.2 group B and 3.1.3 group C) can be better presented in a table.

  • Thank you for this comment. We presented the data in a table at the end of each sub-section to simplify the reading.

4) In the opinion of the authors, can the supplements analyzed in the article be suggested as supplements in the form of drugs for children of this age in the medical routine?

  • Thanks for this question. We believe that these natural compounds can be used to prevent recurrent tonsillitis episodes in children. We have clarified this point in the discussion.

Reviewer 2 Report

The study is interesting form the clinical point of view because it is a conservative and safe proposal for the prevention of recurrences of tonsilitis in children, preventing the use of repeated antibiotics and anti-inflammatory perhaps in the near future.

But the paper must be improved in several aspects mainly in a better and more clear description of the Material and Methods Section in order to clearly compare and reproduce your findings

I send my recommendations

Author Response

Dear Reviewer

Thanks for reviewing our manuscript and for the valuable comments that helped us clarify some relevant aspects that were missed or unclear in the first version of the paper. We have read your comments and have made all the necessary changes to address comments and concerns. We hope that the changes made in the revised manuscript and responses provided below have adequately addressed the reviewers’ comments and made this paper stronger.

The study is interesting form the clinical point of view because it is a conservative and safe proposal for the prevention of recurrences of tonsilitis in children, preventing the use of repeated antibiotics and anti-inflammatory perhaps in the near future. But the paper must be improved in several aspects mainly in a better and more clear description of the Material and Methods Section in order to clearly compare and reproduce your findings

1) The Titer: Is too long and must be changed and  reduced considerably and specified clearly that this study was performed only in children

  • We changed the title by reducing its length and by specifying that the study was conduct on children

2)  The Authors: There are too many. I recommend to suppress at least 2 (from Rome, probably, because they didn´t seem not to participate in the study)

  • We have removed Prof Antonio Greco and Marco de Vincentiis from Rome.

3)  The Abstract: The acronymus of OS must be explained in its first use.

  • Thanks, we have explained the acronymus of OS

4) The dosage and the time of adminsitration of OS in A and B groups, must be specified.

  • We specified these details

5) The conclusion is largely speculative and must be changed completely.

  • We have rewritten the conclusion

6)  The Introduction: You can specifie if this OS is actually marketed in Italy and other countries and give the trade name instead of repeat this long composition many times (Difensil®)

  • Thank you. We followed your suggestion and we used the commercial name of the product.

7) The Material and Methods : The inclusion and exclusion criteria must be better and clearly specified

  • We added details in our inclusion criteria

8) The exact composition of each component of Difensil in solution must be written in order to know for every reader not expert on this subject

  • We specified in details the different composition of Difensil in the two groups

9) The time of administration of the solution in groups A and B must be cleraly done. By instance if the dosage was once daily or twice and the moment of ingestión before the breakfast or with lunch or dinner time

  • We clarified this point, both the dosage and the administration moment.

10) If the patients of GA or GB had fever. They received NSAID sor antibiotics like the CG?

  • We specified that children in GA and GB were treated with 500mg of paracetamol in case of fever and with antibiotics in cases that did not resolve within three days. No corticosteroids were used.

11) The randomly assignation was performed independently for each hospital, or was centraliced and they received the allocation niumber only for one people?

  • Each hospital independently randomized the patients. We clarified it in the materials and methods.

12) At T0 how long the number of tonsilitis how long was measured (1week before or more time) please especify. It would be better to measure during 45 days before inclusion for the basal time estimation, in order to have the same time period for the 3 intervals of the study peiod.

  • Thanks for this remark. The number of tonsillitis was considered in the previous three months. We clarified this in the manuscript.

13) The Results Section: The first paragraph defining the number and characteristis of the included children, must me moved to the previous section (previous od the Statistical analysis). This Section must start in the paragrah 3.1 showing and commenting the results obtained “within the group” comparison

  • Thanks for this suggestion. We have modified the manuscript according to it.

14) The Fig 2 must be moved down at the end of the description the resuts of the Control group, at the line number 163

  • Thanks, we have moved down Figure 2

15) In the same way, the Fig, 3 summarizing the results obtained “between” groups must be moved  at the end of comparison at T2 at the line number 200

  • Thanks, we have moved Figure 3 at the end of the comparison.

16)  The Discussion Section:  In the line 207 on the first paragraph the authors say that the duration of the study was of 6 months. I think that is a mistake, because the duration of the study was only od 3 months, divided in two periods of 45 days each

  • We have corrected this mistake.

17) The commentaries about the ways of action of the different components of the OS are mainly especulatives, because this was not the main objectrive of this study base don clínical observations only.

  • We clarified that the explanation that the component effect was only speculative

18) A comment about the short duration of the study period and the need to continue doing more longer studies woukd be highly recommended

  • We added this in the end of the discussion.

19)  The Conclusions Are also especulative. The composition of the OS must be obviated by the trade name And you must emphasize that is one of the first stydy performe in chidren with this very natural compounds that is safe and without side-effects

  • We have rewritten the conclusions avoiding speculative explanation and we emphasized the novelty of the study

20)  The References ; Are good and enough but they must be corrected folloein the Nutrients style and the year of publication must be moded after the list of authors

  • Thanks, we formatted citations according to the Nutrients style.

Reviewer 3 Report

In this study, the authors assessed the effects of an oral supplement formulation on tonsilitis in children. The study design is reasonable and the conclusions are supported by the data. I do not have any major concerns about the manuscript. Although, authors are encouraged to provide their raw data so that it can contribute as a resource for future meta-analyses.

Author Response

Dear Reviewer

Thanks for reviewing our manuscript and for the valuable comments that helped us clarify some relevant aspects that were missed or unclear in the first version of the paper. We have read your comments and have made all the necessary changes to address comments and concerns. We hope that the changes made in the revised manuscript and responses provided below have adequately addressed the reviewers’ comments and made this paper stronger.

In this study, the authors assessed the effects of an oral supplement formulation on tonsilitis in children. The study design is reasonable and the conclusions are supported by the data. I do not have any major concerns about the manuscript. Although, authors are encouraged to provide their raw data so that it can contribute as a resource for future meta-analyses.

  • Thanks for this comment and for reviewing our manuscript. Although we cannot publicly publish the raw data as supplementary material as the study involved patients under 18 and this is not allowed by our National regimentation, we have added in the results section some tables that can be helpful for future meta-analyses. Furthermore, raw data can be requested to the corresponding author, who can share it with authors/researchers if interested.